# Effectiveness of a Physical Therapeutic Exercise Programme for Caregivers of Dependent Patients: A Pragmatic Randomised Controlled Trial from Spanish Primary Care

**DOI:** 10.3390/ijerph17207359

**Published:** 2020-10-09

**Authors:** Federico Montero-Cuadrado, Miguel Ángel Galán-Martín, Javier Sánchez-Sánchez, Enrique Lluch, Agustín Mayo-Iscar, Ántonio Cuesta-Vargas

**Affiliations:** 1Unit for Active Coping Strategies for Pain in Primary Care, East-Valladolid Primary Care Management, Castilla and Leon Public Health System (Sacyl), 47011 Valladolid, Spain; fmonteroc@saludcastillayleon.es (F.M.-C.); magalanm@saludcastillayleon.es (M.Á.G.-M.); 2Department of Physical Activity and Sports Sciences, University Pontificia of Salamanca, 37002 Salamanca, Spain; jsanchezsa@upsa.es; 3Research Group “Planning and assessment of training and athletic performance”, 37002 Salamanca, Spain; 4Department of Physical Therapy, University of Valencia, 46010 Valencia, Spain; enrique.lluch@uv.es; 5Pain in Motion “International Research Group”, 1090 Brussels, Belgium; 6Department of Human Physiology and Rehabilitation Sciences, Faculty of Physiotherapy, Vrije University Brussels, B-1050 Brussels, Belgium; 7Department of Statistics and Operational Research and IMUVA, University of Valladolid, 47005 Valladolid, Spain; agustin@med.uva.es; 8Department of Physiotherapy, Faculty of Heath Sciences, University of Malaga, 19071 Málaga, Spain; 9Institute of Biomedical Research in Malaga (IBIMA), 29010 Málaga, Spain; 10School of Clinical Science, Faculty of Health Science, Queensland University Technology, Brisbane, QLD 4000, Australia

**Keywords:** female caregivers, therapeutic exercise, public health, quality of life, primary care, physiotherapy

## Abstract

Female family caregivers (FFCs) constitute one of the basic supports of socio-health care for dependence in developed countries. The care provided by FFCs may impact their physical and mental health, negatively affecting their quality of life. In order to alleviate the consequences of providing care on FFCs, the Spanish Public Health System has developed the family caregiver care programme (FCCP) to be applied in primary care (PC) centres. The effectiveness of this programme is limited. To date, the addition of a physical therapeutic exercise (PTE) programme to FCCP has not been evaluated. A randomised multicentre clinical trial was carried out in two PC centres of the Spanish Public Health System. In total, 68 FFCs were recruited. The experimental group (EG) performed the usual FCCP (4 sessions, 6 h) added to a PTE programme (36 sessions in 12 weeks) whereas the control group performed the usual FCCP performed in PC. The experimental treatment improved quality of life (d = 1.17 in physical component summary), subjective burden (d = 2.38), anxiety (d = 1.52), depression (d = 1.37) and health-related physical condition (d = 2.44 in endurance). Differences between the groups (*p* < 0.05) were clinically relevant in favour of the EG. The experimental treatment generates high levels of satisfaction.

## 1. Introduction

In developed countries, due to the sociodemographic and epidemiological changes that have occurred in recent decades, together with an ongoing population ageing, an exponential increase in the need of care is expected for people in situations of dependency [1,2]. This could lead to a crisis in the provision of care to the dependent and a significant impact on available health and social resources in these countries [3]. Spain is one of the countries with the highest levels of ageing [4]. Population projections point out that the percentage of people over 65 years of age, which currently stands at 18.2%, will be 24.9% in 2029 and 38.7% in 2064, and the percentage of people over 80 years old will represent 20.8% of the total elderly population [5]. Living longer implies greater chances of getting sick and suffering from chronic and disabling diseases [6]. In addition, this often causes high dependency in daily activities, leading to higher demands for long-term care [7]. In Spain, most dependent patients live at home and rely on their relatives due to limited hours of available home care, difficulty accessing public services, and, sometimes, resistance to accepting outside help [8]. Previous studies in Spain have shown that the majority of family caregivers are women, who not only spend more hours on caregiving responsibilities but also perform more arduous tasks and receive less help than male caregivers [9,10].

Female family caregivers (FFCs) represent significant savings for care provided by the state’s formal care systems. A US study [11] estimated a total value of informal care of USD 221 billion to USD 642 billion in 2012, which is equivalent to 1.4–4.0% of the US gross domestic product (GDP). In 2008, it was estimated that family caregivers in Spain dedicated around 4600 million hours caring for dependent people, with the value of this family care estimated to range between EUR 18.9 billion and EUR 53.3 billion. These values are equivalent to 29% and 61% of the health expenditure of the National Health System and between 1.7% and 4.9% of the Spanish GDP [12]. FFCs carry out an activity that the health systems would be unable to completely replace, firstly, because the socio-sanitary cost would be practically unaffordable, and secondly, because it would be very difficult to satisfy the affective and emotional demands of the dependent people [13]. For all these reasons, it is easy to understand that FFCs constitute one of the basic supports of socio-health care for dependence in most developed countries.

Caring for a dependent family member often requires exclusive dedication. This means that, in most cases, FFCs have to put their hobbies, work, and even their own health in the background [14]. The scientific literature reflects that FFCs, as a consequence of the care they provide, may suffer significant physical health (tiredness [15], musculoskeletal pain [16], physical deconditioning [17], etc.), mental health (depression [18], stress [19] and anxiety [20], high levels of subjective burden [21] etc.), and social level (social isolation [22], family problems [23], etc.) problems. All these factors negatively affect the quality of life of caregivers [24], with evidence suggesting that caregivers are at increased risk of serious illness and death [25]. FCCs are more frequently affected by all these health problems than male caregivers [26,27] In addition, this situation can also have repercussions for the care recipient. In this sense, the main consequences of this situation are an increase in the use of socio-sanitary resources, a premature institutionalisation and even mistreatment of the dependent person [28].

In order to alleviate the consequences of providing care on FFCs and, therefore, to improve the situation of care for dependent patients, so-called care programmes for caregivers have been developed [29]. In Spain, these programmes for family caregivers are provided by public health services and sometimes patients’ associations. In primary care (PC) of SACYL (public service of Health Castilla y León, Spanish Public Health System), the Family Caregiver Care Programme (FCCP) [30] was developed. It is intended for family caregivers with health problems. Among the most used interventions, family caregiver programmes include mutual support groups or respite interventions, cognitive behavioural therapy, care education programmes, emotional support and self-help groups, and psychological interventions [29]. Several reviews [31,32,33,34,35,36,37,38] have been conducted to determine the effectiveness of family care programmes for caregivers, and results are inconclusive and conflicting. These reviews highlighted that, although the value of conducting family caregiver support interventions is unquestionable, more research on interventions that improve the situation of family caregivers is needed [38]. In this regard, some authors have noted the importance of developing therapeutic strategies to promote the active participation of FFCs [39]. 

Properly performed physical exercise has been found to be one of the best strategies available to promote people’s health and well-being [40]. Despite the demonstrated benefits of physical exercise, few randomised controlled trials (RCTs) [41,42,43,44,45] have evaluated the effects of physical exercise on family caregivers. Most of these studies evaluated FFCs of patients with dementias, and the recruitment process was carried out by patient associations. Moreover, studies were often carried out at caregivers´ home using a face-to-face strategy, and caregivers were supervised by telephone or, in very few cases, through a monitor. In our country, Prieto et al. [46] showed improvement in health-related quality of life (HRQL) and fitness in Spanish FFCs of regional associations for relatives of patients with Alzheimer’s disease after implementing a face-to-face exercise programme at caregivers’ homes. In line with this, several reviews [47,48,49] concluded that physical exercise interventions applied to FFCs improved their stress, depression, subjective burden, quality of life, and physical condition. However, it should be emphasised that these results must be taken with caution due to a lack of many RCTs, a convenience sample selection in many cases, and a lack of detail of the intervention performed. There has also been heterogeneity in these investigations in both the exercise programmes performed and the outcome measures used. Finally, reviews [47,48,49] conclude that it is necessary to conduct more clinical RCTs that have higher methodological quality and include caregivers of family members who present other pathologies, not only dementias. They also highlight that health professionals should consider physical exercise as a possible treatment in the health public system. To date and to our knowledge, RCT-based interventions assessing the effects of physical exercise on caregivers in the Spanish Public Health System are lacking.

The primary aim of this research was to compare the effects of adding a physical therapeutic exercise (PTE) programme to a conventional FCCP on HRQL, psychological well-being, and health-related physical condition (HRPC) in PC FFCs. As a secondary aim, compliance rate and satisfaction with treatment were registered. We hypothesised that FFCs receiving PTE in addition to the FCCP would show more improvements in HRQL, psychological well-being, and HRPC than FFCs only receiving the usual FCCP. Results of this research might contribute to lay the basis for including PTE as a therapeutic tool within the FCCP of Spanish public health care.

## 2. Materials and Methods 

A multicentre RCT was conducted in 2016 in two PC centres of the Valladolid-East Health Area of SACYL (public service of Health of Castile y Leon, Spanish Public Health System). The study was approved by the Ethical Committee for Clinical Research (CEIC) of Valladolid-East Health Area (CEIC: PI-13-88). The trial was registered (ClinicalTrials.gov NTC03675217). The trial was conducted in accordance with the recommendations of the SPIRIT statement [50] and CONSORT statement [51]. In addition, the PRECIS-2 tool was used for the study design [52]. 

### 2.1. Participants

FFCs were recruited from the FCCP of the Pilarica and Circular PC centres (Valladolid, Spain). Both nurses and physicians participated in the recruitment process. They received a two-hour training session where they were informed of the study aims and inclusion criteria. All FFCs from the two basic health zones who could potentially participate in the study were identified. October 1 was established as a starting point for participant’s recruitment. From that date, all FFCs who attended a medical consultation received oral and written information about the study aims and were invited to participate in the study. If they agreed to participate, it was then determined whether they met the inclusion criteria.

Caregivers were included in this study if they were female, had been included in the FCCP, had a direct relationship with the dependent patient and provided continuous care for at least 6 months, were over 50 years of age, and had no changes in medication for at least 3 months prior to assessment.

Exclusion criteria of this study associated pathologies that could prevent caregivers from performing the physical exercise programme (moderate-intensity exercise), participation in another family care programme for caregivers, participation in any regular physical activity programme, and caregivers with support from one or more full-time formal caregivers to care for the dependent relatives. The recruitment process ended when the sample size required by the study was reached.

### 2.2. Sample Size Calculation 

By including 68 individuals in the sample, we can obtain an 80% power to detect differences of two points when testing equality of SF-36 summary of physical health means [53,54,55], with a type I error of 5%. For this calculation, we assume a standard deviation of around 7.46 [53] and an attrition rate of around 15%.

### 2.3. Randomisation

Once the FFCs were selected, randomisation was performed. Each FFC was assigned an alphanumeric code. The alphabetical part of the code corresponded to the participant’s PC centre, while the numerical part was allocated by correlative numbers. FFCs were assigned to a control group (CG) or an experimental group (EG).

### 2.4. Blinding

Members of the research group who performed the assessments did not know the allocated group of FFCs. The intervention common to both groups (FCCP) was applied by PC professionals who also did not know the allocated group of FFCs. The statistician who analysed the results was not in contact with the participants of the study and was blinded to the type of intervention assigned to each group. Due to the characteristics of the intervention, neither the physiotherapist who applied the PTE programme, nor the FFCs who received the intervention could be blinded. 

### 2.5. Interventions

Both groups (experimental and control) carried out the usual FCCP delivered in PC centres (SACYL). In addition, the CG received two phone calls a month for 3 months (six phone calls) to reinforce the content taught in the programme received, while the EG received a PTE programme led by a physiotherapist (Figure 1). All the interventions were performed in the facilities of the PC centres. The adverse effects stemming from the interventions were registered by the staff of the PC team.

#### 2.5.1. Usual Family Caregiver Care Programme

The usual FCCP [30] is intended for family caregivers of dependent patients who have health problems in the community. The aims of this programme are to provide (1) guidance on care, (2) orientation on social resources, and (3) education for groups of caregivers.

In order to homogenise the intervention and avoid bias, professionals who were going to conduct the usual FCCP had several meetings to plan the programme before beginning the intervention. This programme was delivered in groups with a frequency of one session per week over four consecutive weeks (January to February), and each session lasted 90 min. In the first three sessions, participants received only information (theoretical training), whereas the last session was theoretical and practical. The sessions were carried out by a nurse (first and second sessions), a social worker (third session), and a physiotherapist (fourth session). All were PC workers with more than 10 years’ experience in SACYL FCCPs. The contents of the programme are described in Table 1.

At the end of the sessions, recommendations were made to the CG FFCs about the need to make a weekly self-care plan with some of the activities taught. In the last session, FFCs could learn exercises and recommendations for physical activity so that they could carry them out daily. In order to facilitate these recommendations, they received written and audio-visual material. 

#### 2.5.2. Physical Therapeutic Exercise Programme for Experimental Group

Participants of the EG received, as an additional intervention to the conventional FCCP provided by SACYL, a PTE programme. This was a multi-component, functional and neuromotor PTE programme. It consisted of 36 sessions, lasting around 60 min, with a frequency of three sessions per week (12 weeks—February to May). The PTE programme was always carried out in the same order and included the parts described in Table 2.

The PTE was designed by an interdisciplinary team with extensive experience in PTE in PC, based on the literature and considering the conditions of the FFCs. Further details of the content of all the previous sessions were written in a document. The basic structure of the PTE sessions was in accordance with the recommendations of the American College of Sports Medicine (ACSM) [56]. 

All the sessions were conducted by the same PC physiotherapist, who had extensive experience in applying PTE programmes. The sessions were performed in groups and allowed incorporating aimed tasks, dual tasks, and games. As the programme progressed, the difficulty of the physical and cognitive loads increased for the FFCs. Preserving the safety of FFCs during TPE sessions was an objective. It should be noted that although the sessions were developed through group strategy, the trainer paid special attention to the individual needs of each participant, adapting the exercise to their individual condition. The intensity of the aerobic exercise was controlled through the Borg rating of perceived exertion [57]. In the initial stage, the intensity of exercise was maintained between 11 and 13, reaching up to 15 or 16 points on the Borg scale in the later phases. Regular blood pressure checks were carried out to prevent cardio-cerebral risk. 

The TPE programme was conducted in the physiotherapy room of the PC centre, always under the same environmental conditions. For the development of the sessions, the existing material in the physiotherapy unit was used: elastic bands (Theraband^®^), weights, mats, chairs, balls, etc.

### 2.6. Outcome Measures

FFCs were evaluated before the intervention and at week 20 after the beginning of the intervention. The assessment protocol was previously described by the research team, taking into account the scientific literature, and was recorded in writing in a guide for any evaluator to consult in case of doubt. To standardise the assessments, all assessors received a previous course of two hours. Evaluators were health workers from PC centres who did not participate in the study, with experience in evaluating fitness measures and a minimum of 10 years of working experience in the public health system. Each of the FFCs was evaluated in two sessions. In the first session, sociodemographic variables and the characteristics of care provided were registered, which were recorded through a questionnaire prepared for the occasion and their SACYL PC clinical history. In the first session, HRQL and psychological well-being were also evaluated. In the second assessment session, FFCs’ HRPC was evaluated. Each assessment session lasted 50–60 min. Standardised test batteries were selected to measure the HRPC in FFCs, such as the Afisal Battery [58] and Senior Fitness Test [59].

#### 2.6.1. Primary Outcome Measure

To measure the HRQL experienced, the SF-36 V2 Spanish version was used [60]. The SF-36 is a 36-item scale constructed to survey health status and quality of life [61]. This self-completed survey measures eight domains: physical functioning, physical role, bodily pain, general health perceptions, vitality, social functioning, emotional role, and general mental health. It also includes a summary score of mental and physical health based on psychometry. Each domain is transformed directly on a scale from 0 to 100. Higher scores indicate an overall improvement of quality of life [62]. 

#### 2.6.2. Secondary Outcome Measures

##### Psychological Well-Being Variables

**Subjective Burden:** Change in the FFCs’ subjective burden was assessed using the Zarit Carer Burden Interview (ZBI) [63], which is a self-reported questionnaire that evaluates caregiver burden and asks about many commonly reported difficulties associated with the home care situation (i.e., functional, psychological, behavioural, and economic difficulties) [64]. This questionnaire includes a total of 22 Likert-type questions with answers for each ranging from 0 to 4 (total score: 0–88). Higher scores indicate greater caregiver burden. The reliability of the ZBI is high (intraclass correlation coefficient = 0.71) [65,66].

**Family function:** The APGAR test Spanish version was used to assess FFCs’ perceived family function [67]. The acronym APGAR refers to the five components of family function: adaptability, cooperation, development, affectivity, and resolving capacity. The APGAR is a self-reported questionnaire that includes five Likert-type questions, each with three possible answers (never (0 points), sometimes (1 point), and always (2 points)); the total score can range from 0 to 10. Higher scores indicate better perception of family function by FFCs.

**Anxiety:** The Spanish version of the Anxiety Sub-scale of the Goldberg Anxiety and Depression Scale (GADS) was used to assess anxiety [68]. The GADS is very often used in Spanish PC for detecting anxiety for both for health care and epidemiological purposes. The GADS is a self-reported measure and consists of nine dichotomous (yes/no) items measuring anxiety symptoms in adults that had been presented in the last two weeks. Scores range from 0 to 9. A higher score indicates a higher anxiety level. The cut-off point is 4 or more, and the sensitivity and specificity of this scale are 83.1% and 81.8%, respectively [69]. 

**Depression:** The Spanish version of the short-form 15-item version of the Geriatric Depression Scale (GDS) was used to evaluate FFCs’ depression [70]. The GDS is a reliable screening tool for identifying late-life major depression in the PC setting. The GDS V15 is a self-reported questionnaire and consists of 15 dichotomous (yes/no) items measuring depressive symptoms in adults that had been presented in the last week. The total score ranges between 0 and 15, and higher scores indicate worse depressive symptomatology [70]. The GDS is a valid questionnaire (internal consistency, Cronbach’s α = 0.80) [71].

**Pain intensity**: For the intensity of pain, the visual analogue scale (VAS) was used with an unmarked 100 mm line [72].

##### Physical Health Condition Measures

**Body composition study:** A body composition study was used following the anthropometry based on the criteria established by the WHO [73] and the NHANES [74]. The equipment used in the different determinations was previously calibrated following the manufacturers’ standards. Each evaluation was performed three times, and the mean was taken for the analysis. The body density was calculated from the sum of the bicipital, tricipital, subscapular, and suprailiac folds, applying the Durnin and Womersley equations [75]. The Siri formula [76] was used to calculate the percentage of fat mass. Anthropometric assessments were performed by PC workers with ISAK level II.

**Body mass index:** The body mass index (BMI) was calculated from weight (kg) and height (metres), which were combined to report BMI in kg/m^2^. The BMI was calculated using the Quetelet formula [77].

**Waist–hip ratio:** The waist–hip ratio (WHR) was calculated using the circumference of the waist and hip according to the recommendations established by the WHO [78]. Metabolic risk was considered if the WHR was greater than 0.88.

**Handgrip strength:** Handgrip strength was assessed in both hands by using a hand dynamometer (TKK 5401, Tokyo, Japan). The mean value of both hands (kg/m^2^) was considered the outcome [58,79].

**Lower extremity function:** Lower extremity function was evaluated by using the “30-s chair stand test” [59], the one leg stand test [58] and the Timed Up and Go test [80]. In the 30-s chair stand test [59], the patient was asked to stand upright from a standardised chair (0.43 m in height) with arms folded across the chest and to sit back down as many times as possible in 30 s. The best results of two trials (separated by 3 min) were used for analysis [59]. In the one-leg stand test [58], the patient was instructed to stand barefoot on one leg with eyes closed, while the other leg was flexed at knee level and held at the ankle by the hand of the same side of the body. The number of attempts that the subject needed to complete 30 s of the static position was measured and used for analysis. In the Timed Up and Go test [80], the participant rises from a chair, walks 3 m, turns around (circling a cone), and returns to the chair in the shortest possible time. The best score (in seconds) of two attempts was recorded and used for analysis.

**Mobility:** Mobility was assessed using the sit and reach test [81]. The distance between the tips of the fingers from the start to the final position of trunk flexion was recorded. The best result of three trials (expressed in cm) was considered for analysis [82].

**Endurance:** Maximal oxygen uptake (VO_2_ max) was estimated using the 2-km walking test, following the guidelines of the Afisal-Inefc Battery [58]. For the test, a circuit was designed and a stopwatch (portable digital brand Oregon Scientific, model SL929 Hockenheim Nero, Tualatin, OR, USA) with a precision of tenths of a second (0.1 s) was used to measure the execution time. Heart rate monitors (Polar RS800 CX, Polar Electro, Kempele, Finland) were used to measure the heart rate at the end of the test. The FFCs were positioned with comfortable clothes and shoes and asked to walk as fast as possible without running. The time used to travel 2 km as well as the heart rate at the end of the test were recorded. If participants spend more than 22 min to complete the test, it loses its validity. To calculate VO_2_ max, the following formula was used: VO_2_ max (mL·kg·min^−1^) = 116.2 − 2.98 × (time) − 0.11 × (heart rate) − 0.14 (age) − 0.39 × (BMI).

##### Treatment Adherence 

The rate of compliance with the PTE programme was calculated as the percentage of the 36 exercise sessions completed by the participant. The physiotherapist who supervised the exercise sessions recorded the number of sessions each FFC completed. Authors established a minimum of 25 sessions for each FFC. Overall, a minimum follow-up of 80% adherence is recommended to guarantee satisfactory results when implementing physical exercise programmes for adults and the elderly [83].

##### Satisfaction with Treatment

The Client Satisfaction Questionnaire (CSQ-8) was used in the post-intervention assessment to assess participants’ satisfaction with treatment. The CSQ-8 is a self-report tool used to assess satisfaction [84].

### 2.7. Statistical Analysis

Quantitative variables were summarised with means ± standard deviations, and qualitative variables with were summarised with percentages. We calculated 95% confidence intervals (CI95%) for means and percentages. We used the Student’s t-test to test changes in numerical variables within each treatment group and between groups, and we used the chi-squared test for testing differences in percentages. We estimated logistic regression models for predicting successful results in the intervention group. We defined success based on improvement goals in Physical Summary, VAS, and Zarit. In these models, we included variables selected by stepwise procedure, which started with those variables showing p-values lower than 0.15 for the bivariate relation with the corresponding response variable. As a by-product of these analyses, we obtained ORs associated to the factors included in the models and the corresponding ROC curves. A p-value of less than 0.05 was considered statistically significant. Statistical analyses were carried out using the R v3.2.3 statistical package (R Foundation for Statistical Computing, Vienna, Austria). 

## 3. Results

Figure 2 represents the flowchart of this study. A total of 79 FFCs were screened for study eligibility at PC consultations. Seven FCCs were excluded for not fulfilling eligibility criteria, and four declined to participate in the study. Common reasons for declining participation were not having time due to care tasks and being too busy.

At baseline, there were no significant between-group differences in the sociodemographic variables, characteristics of care provided, or clinical variables (all *p* > 0.05). 

The characteristics of the recruited sample are reflected in Table 3. The sociodemographic profile of the FFCs in this study can be summarised as a female caregiver with an average age of 64.35 ± 7.56 years, a relationship of spouse (37.10%) or daughter (48.40%) to the person receiving care, an employment status of retired (40.32%) or without work (27.41%), and a level of academic training of primary studies (64.51%). Regarding the characteristics of care provided, the FFCs had been giving care for an average of 5.35 years, for more than 12 h per day (in 72.58% of cases), and without help or with the help of a family member (67.73%). Care recipients presented an average score of 34.19 ± 21.84 on the Barthel Index (severe dependency). According to this index, 30.64% and 48.38% of the dependent relatives had total and severe dependence for the activities of daily living, respectively. Finally, according to the Pfeiffer questionnaire, the dependent relatives presented significant (41.9%) and moderate (23%) cognitive impairment.

### 3.1. Primary Outcome Measure

Results of the HRQL (SF-36) are presented in Table 4. There were between-group differences in HRQL in favour of the EG (*p* < 0.05). Effect size in the EG was moderate in the mental health subscale and large in bodily pain, general health, vitality, and emotional role subscales of the SF-36 as well as the physical component summary. 

### 3.2. Secondary Outcome Measures

Data related to secondary measures are presented in Table 5. Significant between-group differences were found in all the secondary variables (*p* < 0.05) in favour of the EG, except in the variables of family function, fat-free mass, fat mass, WHR, and waist circumference. Effect size in the EG was moderate for BMI and large for the decrease of subjective burden, anxiety, depression, and pain intensity and increase of handgrip strength, lower extremity function, endurance, and flexibility.

### 3.3. Treatment Adherence

First of all, the high retention and participation of FFCs in the research must be highlighted. High retention and participation of FFCs was found. In particular, a retention of 91.17% of the FCCs was reported (62/68). The FFCs of the EG completed 86.97% of the TPE sessions (i.e., an average of 31.31 ± 2.52 sessions, with a range of 25–36 sessions). The drop-out rate during follow-up was 11.11%. Half of the dropouts were not voluntary (i.e., deaths of care recipients).

### 3.4. Satisfaction 

Both groups (CG and EG) were satisfied with the intervention received. Satisfaction with treatment was higher in the EG (96.87%) than in the CG (83.33%). In EG, we must highlight the high scores obtained in satisfaction with the help received (96.87%) and programme recommendation (yes, definitely: 100%)

No adverse effects were recorded in any of the groups. 

#### Therapeutic Success 

Three versions of therapeutic success were defined based on the improvement of different variables observed after the intervention. These were given by obtaining an increase of more than 2.5 points in the physical summary component (PSC) of the SF-36, a decrease of more than 2.3 points in pain intensity (VAS scale), and a decrease of more than 9 points in the subjective burden (ZBI). For defining therapeutic success, predictive models based on the initial typology of participants were estimated. For each of the three estimated models, scores defined as a linear combination of the explanatory variables with the coefficients shown in Table 6 were created. The transformation of these scores by a function allowed us to obtain estimated probabilities of obtaining success based on explanatory variables.

By establishing cut-off points at 0.5, 0.62, and 0.54, prognostic rules of therapeutic success in the physical summary component of HRQL, pain intensity (VAS), and subjective burden (ZBI) were obtained with estimated sensitivity/specificity values of 77.8%/78.6%, 80%/82.2%, and 58.8%/60%, respectively. 

## 4. Discussion

The results of this study suggest that the implementation of PTE in addition to the usual FCCP produced significant and clinically relevant improvements in the HRQL, psychological well-being, and HRPC of FFCs included in a FCCP in two PC centres of the Spanish Public Health System. In addition to the good clinical results, the adherence and satisfaction of FFCs who received this programme were high, and the intervention had no adverse effects. Our findings may lead to a rethinking of current planning of the FCCPs delivered in health services, since PTE does not require additional resources.

The sociodemographic profile of the FFCs participating in the study was similar to the caregivers’ profile shown in the “EDAD survey” [85] conducted by the Instituto Nacional de Estadistica (INE) and was also similar to profiles seen in most of the research on Spanish PC related to this topic [46,86,87,88,89]. 

In the 20-week follow-up, FFCs belonging to the EG showed an improvement of their HRQL in all dimensions of the SF-36 questionnaire. Our results are partially in line with those reported by Hill et al. [54] and Prieto et al. [53], where a physical exercise programme was carried out at the FFCs’ homes. In agreement with other studies, strength exercises and group physical activity guided by professional are decisive in the improving the perception of psychosocial dimensions related to HRQL [90,91]. This could be explained by the fact that these strategies increase social interaction and the social support dimension, which are considered key elements for emotionally positive ageing [92]. It may be argued that a PTE with playful components is responsible for this improvement in post-intervention quality of life [93].

Regarding mental well-being, the reduction of FCCs‘ subjective burden obtained after the PTE programme is relevant, especially considering the limited results obtained with traditional caregiver interventions [37]. Other authors [44,53] also obtained positive results in subjective burden when applying physical exercise programs monitored by trainers at caregivers’ homes. However, Connel-Janevic [94] and Hill et al. [54] did not find any improvement in subjective burden when using telephone-supervised physical exercise. King et al. [41] concluded that one possible reason explaining the FCCs‘ subjective burden was the low physical condition. The significant reduction in subjective burden observed in the EG could be in accordance with the evidenced relationship between perceived overload and psychological well-being in FFCs of dependent patients that has been established in the literature [48]. The FFCs who performed the PTE programme also showed a significant reduction in anxiety and depression. To the best of our knowledge, Castro et al. [42] are the only ones who have assessed the effects of a home-based exercise programme on caregivers’ anxiety with negative results. Other authors [46,95] showed small improvements in FFCs’ depression with a physical exercise programme delivered in the caregivers’ homes, but with a smaller effect size than that of the present investigation. We chose to include games and dual tasks in the PTE programmes as they are considered to be of crucial importance for achieving positive outcomes in FCCs’ psychological well-being. Regarding family functionality, our search of the scientific literature did not reveal any previous research evaluating the effects of PTE programmes. In our study, the influence of the PTE on the perception of family functionality was small. Other types of interventions, such as psychological therapy and family therapy, could more effectively influence family functionality and could be combined with exercise programmes. 

Despite the fact that several studies showed that FFCs have worse physical condition than non-caregivers, most RTCs have primarily considered psychosocial outcomes, and few studies have examined physical function [53,96,97]. Based on our results, we could argue that adding a 12-week PTE programme to the usual FCCP may result in positive effects on most components of the FFCs’ HRPC. These results are in line with those obtained when applying home-based programmes [53,96]. Improving physical condition is essential for FFCs to provide effective care. Interestingly, our study is the first to have evaluated the FFCs’ body composition by anthropometry. Decreases in BMI, fat mass, waist circumference, and WHR were observed at the 20-week follow-up. These changes may be considered a positive contribution to FFCs’ health, due to the relationship of these indicators with metabolic health [98]. The FFCs of the EG improved their handgrip strength and endurance, and it is known that poor strength and endurance are highly correlated with poorer HQRL and adverse health outcomes, including disability and increased hospitalisations, in older adults [99]. 

We were surprised by the CG results (equal to or worse than baseline status). The type of intervention which was implemented could be a factor that explains this lack of improvement. Traditional interventions (e.g., care education, respite programmes, and drugs) have demonstrated limited impact on caregivers’ burden reduction and psychological well-being. In addition, the fact that the intervention for the CG only included four face-to-face sessions (once a week) could also explain these results [100]. Furthermore, Cuevas-Fernández [101] highlighted, in his longitudinal study carried out on Spanish PC, that the passage of time (at 3 and 6 months) while providing care may increase the FFCs’ subjective burden perception. Finally, it could be argued that passive interventions may be less effective than active coping strategies to improve the FFCs’ health status.

In our research, high adherence to the PTE must be highlighted. Despite the fact that some authors [102] consider that physical exercise programmes have more adherence if they are performed individually at the caregivers’ homes, our compliance is similar to or even greater than that obtained by other studies [42,43,44,45] using home-based exercise programmes. Furthermore, the monitoring by a professional increases adherence to exercise sessions [103], and this could have influenced the results. To date, most physical exercise programmes targeting FFCs have been telephone-supervised. The presential monitoring sessions used in our study allow individual adjustment of the treatment in each session based on FFCs’ individual needs (especially for people with pain) and have been shown to be more effective than telephone-supervised monitoring [104]. The fact that the PTE programme was implemented outside the family environment could be another factor influencing adherence. FFCs were able to perform the exercise without care interruptions, while at the same time it was a moment of care-break or distraction [105]. The group-based nature of the EG intervention favoured socialisation [106,107] and may have taken many FFCs out of their social isolation.

### 4.1. Limitations and Strengths 

The main limitation of this study was the difference in the intervention burden load in favour of the EG. This may have influenced our results due to the potentially stronger therapist–patient relationship established in the EG. Even so, the intervention received by the CG was the usual care that is used in FCCP of SACYL PC centres. Another limitation is that follow-up was only done until week 20. A longer-term follow-up would be desirable for future research applying the proposed experimental treatment.

The strengths of the study include the pragmatism of the intervention (Appendix A: Figure A1) and results obtained. It should be noted that the intervention was carried out within the usual health care setting and had a marked pragmatic component. In addition, the important results obtained in this study in terms of FFCs’ anxiety and subjective burden contrast with limited results obtained in these variables with the traditional caregiver care programmes. Furthermore, using a group intervention and monitoring from PC physiotherapy, we achieved similar or superior (in some variables) results when compared to those of other studies [55,101] that followed an individual strategy at caregivers’ homes with monitoring by a trainer. The group intervention represents a strategy for saving both economic and human resources, which makes the development of this strategy feasible for PC. Considering the currently available human resources in Spain, it would be impossible to deliver home exercise programmes for each of the FFCs. Furthermore, it has been shown that if caregivers have a better physical condition, they reduce the consumption of social and health resources [101], and positive effects can even be translated to the dependent relatives.

### 4.2. Practical Implications and Recommendations for Research

The objective of this intervention was to improve the conditions of FFCs, who perform essential and irreplaceable work in the care of dependent patients. The health and well-being of FCCs are of vital importance, since nearly 1.5 million caregivers in Spain are currently providing informal care for a dependent family member, and approximately one-third of that number are reported to experience a negative impact on their health and quality of life [21]. This is more relevant considering that the number of informal care users is expected to rise significantly in the coming years. In particular, the number of informal caregivers in Spain is expected to experience an increase of 140%, from approximately 1.2 million in 2010 to more than 2.8 million in 2060 [108]. To the best of our knowledge, the present study is the first RCT to evaluate pragmatically the effects of a PTE programme for FFCs of dependent patients in PC in the public health system. Based on the results of this study, we believe that FFCs should receive the health care they need in PC. On the other hand, some FFCs do not respond to these types of interventions. In that case, more individual treatments or a combination of the TPE programme with drugs, psychological therapies, etc. may be needed. 

It should be noted that the FCCP of SACYL only includes caregivers who have health problems. It would be interesting, taking into account the preventive properties of exercise [109], to conduct further research to determine whether PTE could be a useful preventive tool to avoid negative repercussions for the FFCs as a consequence of providing care. It would also be interesting to evaluate the long-term effects of the current intervention and to evaluate the cost-effectiveness of the implementation of this type of programme in FCCPs of the public health system. We developed a series of clinical prediction rules to determine what types of FFCs can benefit from these programs. From a clinical point of view, this would be interesting, since physicians would have tools that would allow them to refer the caregivers to FCCPs with PTE programmes, knowing in advance that the programme may help them. It would be interesting if these rules were corroborated in future research with a larger sample size.

## 5. Conclusions

The implementation of a PTE programme within the usual FCCP improves the mental well-being, HRPC, and HQRL of a group of FFCs in PC. In addition, the compliance and satisfaction with the PTE programme were high, and there was a lack of adverse effects. FFCs should have access to PTE programmes as part of caregiver care programmes in PC. Health managers should consider implementing PTE programmes within FCCPs of PC, considering the important benefits they produce.

## Figures and Tables

**Figure 1 ijerph-17-07359-f001:**
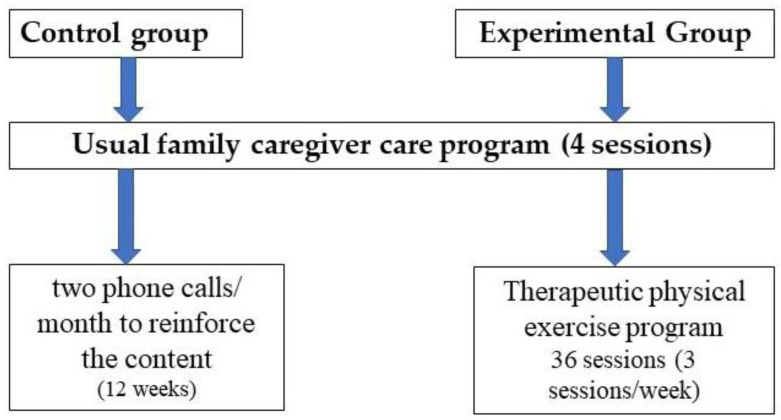
Study arms.

**Figure 2 ijerph-17-07359-f002:**
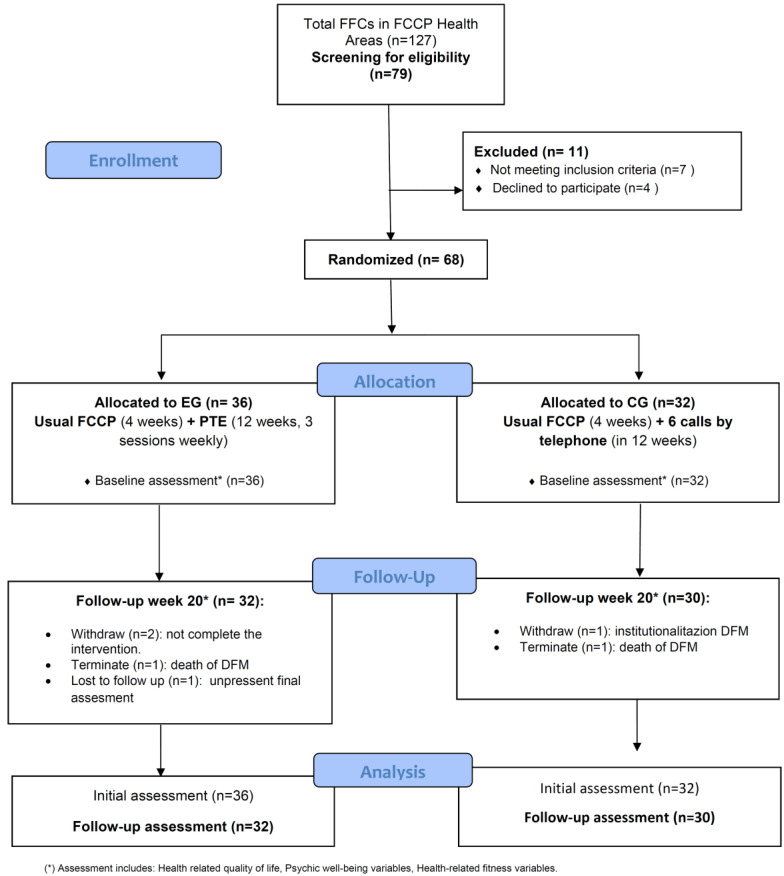
Flowchart of the study.

**Table 1 ijerph-17-07359-t001:** Usual family caregiver care programme.

Usual Family Caregiver Care Programme (6 h)	
First session (90 min)	‘Knowledge of a family caregiver’: provide care and take care of yourself. Day-to-day care: adaptation of homes to the needs of dependent people, use of care aids, and promotion of the skills and self-esteem of the dependent. Caring for daily activities: food, hygiene and bathing, mobility, prevention of falls, incontinence, inactivity and sadness, sleep, and postural changes. Difficult care problems: agitation, aggression, anxiety, and anger.
Second session (90 min)	‘Who cares for the caregiver?’: Effects and consequences of care on the health of the family caregiver. Caregiver syndrome: physical and psychosocial manifestations. Warning signs: fatigue, sleep problems, anxiety, depression, cognitive failure, accidents, focus only on care, changes in appetite, routine actions, apathy, and excessive consumption. Dysfunctional thoughts: anger, sadness, and guilt.
Third session (90 min)	‘Prevention measures and coping strategies’: Family caregiver rights. Dependence resources. Planning the future: foresight, knowledge, and knowing how to ask for help. Communication. Facing a situation: recognising feelings, evaluating feelings, and making a decision. Relieve emotional overload. Search for well-being: healthy behaviours. Importance of own time and self-care plan. Relaxation.
Fourth session (90 min)	‘The care of our body’: Importance of an active and healthy lifestyle. Ergonomics and postural hygiene to prevent back pain. Mobilisation and transfer of dependent patients. Self-care activities that promote well-being and prevent musculoskeletal pain: stretching, breathing exercises, relaxation, and physical exercise.

**Table 2 ijerph-17-07359-t002:** Session structure of physical therapeutic exercise.

Session Structure of Physical Therapeutic Exercise	
Warm-up phase (10 min)	Exercise of aerobic activation and active joint mobilisation.
Main part (35–40 min)	Preventive exercises: Coordination; Balance; Agility exercise. Strength exercises (gym weights, and own body weight). Cardiovascular resistance (aerobic exercise): First four weeks: aerobic exercises at 60–65% of maximal heart rate (HRmax); Following weeks: intensity increased to 70–80% (HR max), which means about 3 to 6 metabolic equivalents.
Cooling-down phase (10–15 min)	Exercises to reach a state of calm: Breathing work; Stretching; Relaxation.

**Table 3 ijerph-17-07359-t003:** Sociodemographic profile and characteristics of care provided.

	Total Sample (*n* = 62)	CG (*n* = 30)	EG (*n* = 32)
**Sociodemographic information**			
Age (mean ± SD)	64.35 ± 7.56	63.77 ± 7.83	64.91 ± 7.38
Marital status (%)			
Single	16.12	20.00	12.5
Married/in couple	66.12	60.00	71.9
Divorced	9.67	13.33	6.25
Widow	8.06	6.66	9.37
Education level (%)			
Incomplete Primary education	8.06	6.66	9.37
Primary education	64.51	63.33	65.62
Secondary education	12.9	16.66	9.37
University degree	14.52	13.33	15.62
Employed situation (%):			
Pensioner	40.32	36.66	43.75
Active	14.52	16.66	12.50
Unemployed	17.74	23.33	12.50
Home chores	27.41	23.33	31.25
Relationship with care recipient			
Daughter	48.38	46.66	50
Wife/woman couple	37.09	40.00	34.37
Mother	4.83	3.33	6.25
Sister	4.83	6.66	3.12
Daughter in law	4.83	6.25	3.33
Pain			
Lower back pain	69.35	66.66	71.87
Neck pain	59.68	56.66	62.50
**Characteristics of care provided**			
Caregiving duration (age. mean ± SD)	5.35 ± 3.75	4.57 ± 3.08	6.09 ± 4.21
Hours per day providing care (mean ± SD)			
1 to 12	27.41	29.99	25.00
>12	72.58	70.00	75.00
Help for providing care			
No	25.80	23.33	28.12
Yes, from a family member	41.93	40.00	43.75
Yes, from a family member who does not live in the family home	17.74	20.00	15.62
Yes, from a formal caregiver	14.52	16.66	12.50
Functional status of dependent family member (Barthel Index) (mean ± SD)	34.19 ± 21.84	35 ± 23.16	33.34 ± 20.88
Cognitive status of dependent family member (Pfeiffer Index) (mean ± SD)	5.97 ± 2.73	5.8 ± 2.69	6.15 ± 2.77

**Table 4 ijerph-17-07359-t004:** Health-related quality of life (HRQL) outcomes.

HRQL (SF-36)	Group CG (*n* = 30) EG (*n* = 32)	Pre	Post	Intra. Dif (IC 95%)	Inter. Dif (IC 95%)	Effect Size
Physical Function (FF)	CG	72.67 ± 18.08	72.83 ± 17.80	−0.16 (−2.09 to 1.76)	2.33	0.49
	EG	70.15 ± 14.34	72.65 ± 13.38	−2.50 (−4.01 to −0.98) *	(−0.07 to 4.74)	
Physical Role (RF)	CG	66.67± 28.11	65.83 ± 28.22	0.83 (−5.87 to 7.54)	7.86	0.44
	EG	64.84 ± 22.77	71.87 ± 22.67	**−7.03 (−13.18 to −0.87) ***	(−1.05 to 16.78)	
Bodily Pain (BP)	CG	47.41 ± 19.84	44.91 ± 16.13	2.50 (−0.74 to 5.74)	12.03	**1.01**
	EG	48.20 ± 17.02	57.73 ± 15.11	−9.53 (−14.68 to −4.37) **	(6.04 to 18.02) ^‡^	
Mental Health (MH)	CG	53.17 ± 19.93	49.67 ± 18.93	3.50 (1.35 to 5.64) *	11.21	**1.69**
	EG	51.56 ± 13.76	59.28 ± 14.35	−7.71 (−10.39 to −5.04) **	(7.85 to 14.57) ^‡^	
Vitality (VT)	CG	52.50 ± 21.45	49.50 ± 19.04	3.00 (0.20 to 5.79) *	11.82	**1.52**
	EG	56.33 ± 19.13	65.15 ± 16.83	−8.82 (−11.71 to −5.94) **	(7.89 to 15.76) ^‡^	
Social Functioning (SF)	CG	66.17 ± 25.88	65.66 ± 22.85	0.50 (−2.50 to 3.50)	1.91	0.20
	EG	61.641 ± 9.28	63.04 ± 16.89	−1.41 (−5.25 to 2.43)	(−2.87 to 6.69)	
Emotional Role (RE)	CG	63.33 ± 34.29	62.22 ± 28.68	1.11 (−7.21 to 9.43)	2.15	**0.98**
	EG	62.49 ± 27.76	63.53 ± 25.90	−1.04 (−8.81 to 6.72)	(−8.99 to 13.30)	
Mental Health (MH)	CG	54.60 ± 22.09	55.13 ± 21.64	−0.53 (−2.51 to 1.44)	2.59	0.53
	EG	58.12 ± 16.94	61.25 ± 16.17	−3.12 (−4.71 to −1.54) **	(0.10 to 5.07) ^†^	
Physical Component Summary (PCS)	CG	48.49 ± 6.52	47.73 ± 5.67	0.75 (−0.361 to 1.87)	3.38	**1.17**
	EG	47.57 ± 4.65	50.20 ± 4.71	−2.63 (−3.62 to −1.63) **	(1.92 to 4.85) ^‡^	
Mental Component Summary (MCS)	CG	43.47 ± 10.64	43.26 ± 9.11	0.20 (−1.29 to 1.70)	1.27	0.36
	EG	44.51 ± 8.27	45.59 ± 7.22	−1.07 (−2.10 to −0.04) *	(−0.50 to 3.06)	

Data are expressed as mean ± standard deviation; CG, control group; EG, experimental group; Pre, values before intervention; Post, values at week 20; Intra. Dif, intragroup difference; Inter. Dif, intergroup difference; * and ** indicate significant intragroup differences (*p* < 0.05 and *p* < 0.01, respectively); ^†^ and ^‡^ indicate significant intergroup differences (*p* < 0.05 and *p* < 0.01, respectively); Values in bold indicate a large effect size (>±0.8)

**Table 5 ijerph-17-07359-t005:** Secondary outcomes.

	Group CG (*n* = 30) EG (*n* = 32)	Pre	Post	Intra. Dif (IC 95%)	Inter. Dif (IC 95%)	Effect Size
Psychological well-being						
Subjective Burden (ZBI)	CG	54.93 ± 15.40	56.67 ± 15.13	−1.73 (−2.55 to –0.91) **	−10.32	**−2.38**
	EG	57.06 ± 13.45	48.46 ± 11.19	8.59 (6.539 to 10.64) **	(−12.52 to −8.13) ^‡^	
Family Function (Apgar)	CG	7.06 ± 2.34	7.16 ± 2.42	−0.10 (−0.32 to 0.12)	0.15	0.25
	EG	7.22 ± 1.89	7.47 ± 1.65	−0.25 (−0.45 to −0.04) *	(−0.15 to 0.45)	
Anxiety (GADS)	CG	4.30 ± 2.25	4.47 ± 2.24	−0.17 (−0.51 to 0.17)	−1.57	**−1.52**
	EG	4.53 ± 2.01	3.12 ± 2.13	1.40 (0.99 to 1.81) **	(−2.09 to −1.05) ^‡^	
Depression (GDS)	CG	5.47 ± 2.69	5.77 ± 2.82	−0.30 (−0.61 to 0.01)	−1.80	**−1.37**
	EG	5.91 ± 2.66	4.41 ± 1.66	1.50 (0.90 to 2.09) **	(−2.46 to −1.13) ^‡^	
Pain intensity (VAS)	CG	60.70 ± 12.80	63.01 ± 12.40	−2.3 (−0.54 to 0.09)	−25.7	**−2.66**
	EG	63.20 ± 10.70	39.80 ± 12.80	23.4 (19.5 to 27.3) **	(−30.6 to −20.81) ^‡^	
Health-related physical condition						
BMI (kg/m^2^)	CG	29.28 ± 6.27	29.35 ± 6.38	−0.07 (−0.25 to 0.11)	−0.33	−0.58
	EG	27.53 ± 4.93	27.26 ± 4.59	0.26 (0.03 to 0.49) *	(−0.62 to −0.04) †	
Waist−Hip Ratio	CG	0.85 ± 0.06	0.84 ± 0.08	0.01 (−0.01 to 0.02)	−0.2	−0.24
	EG	0.86 ± 0.07	0.83 ± 0.07	0.03 (0.01 to 0.03) *	(−0.03 to 0.01)	
Waist Circumference (cm)	CG	88.53 ± 14.48	88.36 ± 14.57	0.17(−0.98 to 1.32)	−1.45	− 0.45
	EG	86.01 ± 12.01	84.39 ± 10.82	1.62 (0.41 to 2.84) *	(−3.09 to 0.18)	
Fat free mass (%) (anthropometry)	CG	60.42 ± 6.90	60.27 ± 6.55	0.15 (−0.34 to 0.64)	0.66	0.29
	EG	60.61 ± 5.70	61.42 ± 4.85	−0.81 (−2.14 to 1.03)	(−0.94 to 2.36)	
Fat mass (%) (anthropometry)	CG	39.57 ± 6.09	39.72 ± 6.55	−0.15 (−0.64 to 0.34)	−0.66	−0.29
	EG	39.08 ± 5.70	38.57 ± 4.85	0.51 (−1.03 to 2.14)	(−2.36 to 0.94)	
Handgrip Strength (kg·m^2^)	CG	41.43 ± 6.76	40.74 ± 6.88	0.69 (0.37 to 1.05) *	3.43	**2.23**
	EG	39.92 ± 7.42	42.63 ± 7.67	−2.71 (−3.42 to−2.01) **	(4.21 to 2.65) ^‡^	
Lower extremity function:						
A: Chair stand test	CG	12.46 ± 3.14	11.69 ± 3.12	0.77 (0.27 to 1.26) *	3.48	**2.21**
	EG	10.5 ± 2.19	13.21 ± 2.28	−2.71 (−3.36 to −2.07) **	(2.69 to 4.28) ^‡^	
B.1: One leg stand test, right leg	CG	10.40 ± 4.75	10.47 ± 4.67	−0.06 (−1.01 to 0.88)	−3.53	**−1.01**
	EG	10.88 ± 5.32	7.41 ± 4.68	3.47 (1.95 to 4.98) **	(−5.29 to −1.78) ^‡^	
B.2: One leg stand test, left leg	CG	10.87 ± 4.80	10.7 ± 4.68	0.16 (−0.84 to 1.17)	−4.02	**−1.08**
	EG	11.66 ± 5.42	7.47 ± 5.40	4.18 (2.57 to 5.80) **	(−5.89 to −2.15) ^‡^	
C: Timed up and go test (s)	CG	6.385 ± 1.37	6.85 ± 1.42	−0.46 (−0.72 to −0.21) *	−1.43	**−1.98**
	EG	6.63 ± 1.14	5.66 ± 0.96	0.96 (0.69 to 1.24) **	(−1.80 to −1.07) ^‡^	
Endurance VO_2_ max. (mL·kg·min^−1^)	CG	23.29 ± 5.31	22.57 ± 5.78	0.72 (0.26 to 1.19) *	3.85	**2.44**
	EG	22.89 ± 3.98	26.01 ± 4.40	−3.12 (−3.79 to −2.46) **	(3.05 to 4.65) ^‡^	
Flexibility (sit and reach test) (cm)	CG	12.78 ± 3.38	12.55 ± 3.39	0.23 (−0.40 to 0.84)	3.41	**1.63**
	EG	13.25 ± 4.40	16.44 ± 5.11	−3.18 (−4.06 to −2.31) **	(2.36 to 4.46) ^‡^	

Data are expressed as mean ± standard deviation; CG, control group; EG, experimental group; Pre, values before intervention; Post, values at week 20; Intra. Dif, intragroup difference; Inter. Dif, intergroup difference; * and ** indicate significant intragroup differences (*p* < 0.05 and *p* < 0.01, respectively); ^†^ and ^‡^ indicate significant intergroup differences (*p* < 0.05 and *p* < 0.01, respectively); Values in bold indicate a large effect size (>±0.8)

**Table 6 ijerph-17-07359-t006:** Therapeutic success.

	Physical Health Success (PSC SF−36)				Pain Intensity Success (VAS)				Subjective Burden Success (ZBI)		
	**Coef. 1**	**SE1**	**p1**		**Coef. 2**	SE2	p2		Coef. 3	SE3	p3
Independent term	−2.98	1.13	0.008	Independent Term	5.83	2.18	0.007	Independent term	−8.64	4.5	0.050
Subjective burden (Zarit)	2.08	0.069	0.003	Lower back pain (pain charts)	3.15	1.35	0.020	Physical health (PSC SF−36)	−0.18	0.09	0.051
				Balance (one leg stand test)	0.34	0.14	0.011				

Coef: Coefficient; SE: standar error; *p*: p-value.

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
