# Peer review of "Effectiveness of a Physical Therapeutic Exercise Programme for Caregivers of Dependent Patients: A Pragmatic Randomised Controlled Trial from Spanish Primary Care"

_ijerph, 2020, doi:10.3390/ijerph17207359_

Round 1

Reviewer 1 Report

This article deals with a very important subject of physical and psychical condition of caregivers of dependent patients.

Primarily, it offers a proposition to reduce the negative impact of such strenuous work.

If the difference between the tasks for each group was evident, please explain why mobilization of CG was only twice a month, while the sessions of EG were three times a week. Perhaps (it is not known) more frequent mobilization of the control group would improve its results.

Author Response

This article deals with a very important subject of physical and psychical condition of caregivers of dependent patients.

Primarily, it offers a proposition to reduce the negative impact of such strenuous work.

If the difference between the tasks for each group was evident, please explain why mobilization of CG was only twice a month, while the sessions of EG were three times a week. Perhaps (it is not known) more frequent mobilization of the control group would improve its results.

Authors:

Thank you for your comments.

Due to the strong pragmatic component of our clinical trial the usual care program for dependent patients has not been modified, and we are aware that the burden in both groups is very different, but the main objective of our work was to assess the effect of adding a physical exercise program to the usual care program that is carried out in the public health system, and to assess its effect on physical and mental health of caregivers.

Reviewer 2 Report

This is a very nicely done study, and I have little complaints or concerns with the design or interpretation of the results. My main complaint at the present time has to do with English language usage. The authors will need to have good quality English language editing performed on the entire paper, at which time a second review can be performed. 

The attached file indicates problems with usage and other English language issues in the first few pages, and I leave it to the authors to locate and correct other instances.

Author Response

R2:

This is a very nicely done study, and I have little complaints or concerns with the design or interpretation of the results. My main complaint at the present time has to do with English language usage. The authors will need to have good quality English language editing performed on the entire paper, at which time a second review can be performed. 

The attached file indicates problems with usage and other English language issues in the first few pages, and I leave it to the authors to locate and correct other instances.

Authors:

Dear revisor, thank you for your comments.

A revision of the text has been carried out to improve the use of the English language, and the text has been submitted to a professional service of Proof-Reading Service. We are confident that we have substantially improved the quality of the text.

Round 2

Reviewer 2 Report

Looks good!

Author Response

Dear Reviewer, we adjust the tables following the indications. Thanks for your feedback end review